# Which Space Partitioning Tree to Use for Search?

**P. Ram**
Georgia Tech. / Skytree, Inc.
Atlanta, GA 30308
p.ram@gatech.edu

**A. G. Gray**
Georgia Tech.
Atlanta, GA 30308
agray@cc.gatech.edu

## Abstract

We consider the task of nearest-neighbor search with the class of binary-space-partitioning trees, which includes *kd*-trees, principal axis trees and random projection trees, and try to rigorously answer the question *"which tree to use for nearest-neighbor search?"* To this end, we present the theoretical results which imply that trees with better vector quantization performance have better search performance guarantees. We also explore another factor affecting the search performance – margins of the partitions in these trees. We demonstrate, both theoretically and empirically, that large margin partitions can improve tree search performance.

## 1 Nearest-neighbor search

Nearest-neighbor search is ubiquitous in computer science. Several techniques exist for nearest-neighbor search, but most algorithms can be categorized into two following groups based on the indexing scheme used – (1) search with hierarchical tree indices, or (2) search with hash-based indices. Although multidimensional binary space-partitioning trees (or BSP-trees), such as *kd*-trees [1], are widely used for nearest-neighbor search, it is believed that their performances degrade with increasing dimensions. Standard worst-case analyses of search with BSP-trees in high dimensions usually lead to trivial guarantees (such as, an $\Omega(n)$ search time guarantee for a single nearest-neighbor query in a set of $n$ points). This is generally attributed to the "curse of dimensionality" – in the worst case, the high dimensionality can force the search algorithm to visit every node in the BSP-tree.

However, these BSP-trees are very simple and intuitive, and still used in practice with success. The occasional favorable performances of BSP-trees in high dimensions are attributed to the low "intrinsic" dimensionality of real data. However, no clear relationship between the BSP-tree search performance and the intrinsic data properties is known. *We present theoretical results which link the search performance of BSP-trees to properties of the data and the tree*. This allows us to identify implicit factors influencing BSP-tree search performance — knowing these driving factors allows us to develop successful heuristics for BSP-trees with improved search performance.

Each node in a BSP-tree represents a region of the space and each non-leaf node has a left and right child representing a disjoint partition of this region with some separating hyperplane and threshold $(\mathbf{w}, b)$. A search query on this tree is usually answered with a depth-first branch-and-bound algorithm. Algorithm 1 presents a simplified version where a search query is answered with a small set of neighbor candidates of any desired size by performing a greedy depth-first tree traversal to a specified depth. This is known as *defeatist tree search*. We are not aware of any data-dependent analysis of the quality of the results from defeatist BSP-tree search. However, Verma et al. (2009) [2] presented adaptive data-dependent analyses of some BSP-trees for the task of vector quantization. These results show precise connections between the quantization performance of the BSP-trees and certain properties of the data (we will present these data properties in Section 2).

**Algorithm 1** BSP-tree search

**Input:** BSP-tree $T$ on set $S$,
         Query $q$, Desired depth $l$
**Output:** Candidate neighbor $p$

current tree depth $l_c \leftarrow 0$
current tree node $T_c \leftarrow T$
**while** $l_c < l$ **do**
    **if** $\langle T_c.\mathbf{w}, q \rangle + T_c.b \leq 0$ **then**
        $T_c \leftarrow T_c.\text{left\_child}$
    **else**
        $T_c \leftarrow T_c.\text{right\_child}$
    **end if**
    Increment depth $l_c \leftarrow l_c + 1$
**end while**
$p \leftarrow \arg\min_{r \in T_c \cap S} \|q - r\|.$

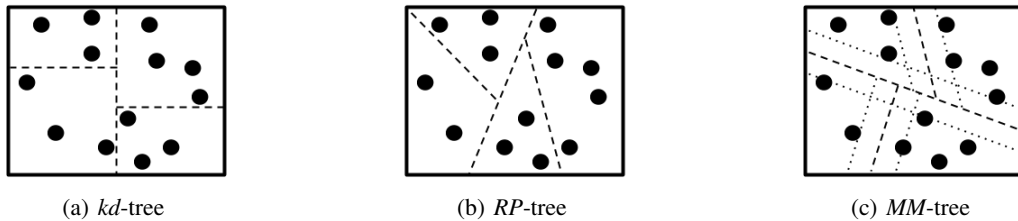

(a) *kd*-tree        (b) *RP*-tree        (c) *MM*-tree

Figure 1: **Binary space-partitioning trees.**

We establish search performance guarantees for BSP-trees by linking their nearest-neighbor performance to their vector quantization performance and utilizing the recent guarantees on the BSP-tree vector quantization. *Our results provide theoretical evidence, for the first time, that better quantization performance implies better search performance*[1]. These results also motivate the use of *large margin BSP-trees*, trees that hierarchically partition the data with a large (geometric) margin, for better nearest-neighbor search performance. After discussing some existing literature on nearest-neighbor search and vector quantization in Section 2, we discuss our following contributions:

- We present performance guarantees for Algorithm 1 in Section 3, linking search performance to vector quantization performance. Specifically, we show that for any balanced BSP-tree and a depth $l$, under some conditions, the worst-case search error incurred by the neighbor candidate returned by Algorithm 1 is proportional to a factor which is

$$O\left(\frac{2^{l/2}\exp(-l/2\beta)}{(n/2^l)^{1/O(\mathbf{d})}-2}\right),$$

where $\beta$ corresponds to the quantization performance of the tree (smaller $\beta$ implies smaller quantization error) and $\mathbf{d}$ is closely related to the doubling dimension of the dataset (as opposed to the ambient dimension $\mathbf{D}$ of the dataset). This implies that better quantization produces better worst-case search results. Moreover, this result implies that smaller $l$ produces improved worst-case performance (smaller $l$ does imply more computation, hence it is intuitive to expect less error at the cost of computation). Finally, there is also the expected dependence on the intrinsic dimensionality $\mathbf{d}$ – increasing $\mathbf{d}$ implies deteriorating worst-case performance. The theoretical results are empirically verified in this section as well.

- In Section 3, we also show that the worst-case search error for Algorithm 1 with a BSP-tree $T$ is proportional to $(1/\gamma)$ where $\gamma$ is the smallest margin size of all the partitions in $T$.

- We present the quantization performance guarantee of a large margin BSP tree in Section 4.

These results indicate that for a given dataset, the best BSP-tree for search is the one with the best combination of low quantization error and large partition margins. We conclude with this insight and related unanswered questions in Section 5.

## 2   Search and vector quantization

Binary space-partitioning trees (or BSP-trees) are hierarchical data structures providing a multi-resolution view of the dataset indexed. There are several space-partitioning heuristics for a BSP-tree construction. A tree is constructed by recursively applying a heuristic partition. The most popular *kd*-tree uses axis-aligned partitions (Figure 1(a)), often employing a median split along the coordinate axis of the data in the tree node with the largest spread. The *principal-axis tree* (*PA*-tree) partitions the space at each node at the median along the principal eigenvector of the covariance matrix of the data in that node [3, 4]. Another heuristic partitions the space based on a 2-means clustering of the data in the node to form the *two-means tree* (*2M*-tree) [5, 6]. The *random-projection tree* (*RP*-tree) partitions the space by projecting the data along a random standard normal direction and choosing an appropriate splitting threshold [7] (Figure 1(b)). The *max-margin tree* (*MM*-tree) is built by recursively employing large margin partitions of the data [8] (Figure 1(c)). The unsupervised large margin splits are usually performed using max-margin clustering techniques [9].

**Search.** Nearest-neighbor search with a BSP-tree usually involves a depth-first branch-and-bound algorithm which guarantees the search approximation (exact search is a special case of approximate search with zero approximation) by a depth-first traversal of the tree followed by a backtrack up the tree as required. This makes the tree traversal unpredictable leading to trivial worst-case runtime

guarantees. On the other hand, *locality-sensitive hashing* [10] based methods approach search in a different way. After indexing the dataset into hash tables, a query is answered by selecting candidate points from these hash tables. The candidate set size implies the worst-case search time bound. The hash table construction guarantees the set size and search approximation. Algorithm 1 uses a BSP-tree to select a candidate set for a query with defeatist tree search. For a balanced tree on $n$ points, the candidate set size at depth $l$ is $n/2^l$ and the search runtime is $O(l + n/2^l)$, with $l \leq \log_2 n$. For any choice of the depth $l$, we present the first approximation guarantee for this search process.

Defeatist BSP-tree search has been explored with the *spill tree* [11], a binary tree with overlapping sibling nodes unlike the disjoint nodes in the usual BSP-tree. The search involves selecting the candidates in (all) the leaf node(s) which contain the query. The level of overlap guarantees the search approximation, but this search method lacks any rigorous runtime guarantee; it is hard to bound the number of leaf nodes that might contain any given query. Dasgupta & Sinha (2013) [12] show that the probability of finding the exact nearest neighbor with defeatist search on certain randomized partition trees (randomized spill trees and *RP*-trees being among them) is directly proportional to the *relative contrast* of the search task [13], a recently proposed quantity which characterizes the difficulty of a search problem (lower relative contrast makes exact search harder).

**Vector Quantization.** Recent work by Verma et al., 2009 [2] has established theoretical guarantees for some of these BSP-trees for the task of vector quantization. Given a set of points $S \subset \mathbb{R}^\mathbf{D}$ of $n$ points, the task of vector quantization is to generate a set of points $M \subset \mathbb{R}^\mathbf{D}$ of size $k \ll n$ with low average quantization error. The optimal quantizer for any region $A$ is given by the mean $\mu(A)$ of the data points lying in that region. The quantization error of the region $A$ is then given by

$$\mathcal{V}_S(A) = \frac{1}{|A \cap S|} \sum_{x \in A \cap S} \|x - \mu(A)\|_2^2, \tag{1}$$

and the average quantization error of a disjoint partition of region $A$ into $A_l$ and $A_r$ is given by:

$$\mathcal{V}_S(\{A_l, A_r\}) = \left(|A_l \cap S|\mathcal{V}_S(A_l) + |A_r \cap S|\mathcal{V}_S(A_r)\right)/|A \cap S|. \tag{2}$$

Tree-based *structured* vector quantization is used for efficient vector quantization – a BSP-tree of depth $\log_2 k$ partitions the space containing $S$ into $k$ disjoint regions to produce a $k$-quantization of $S$. The theoretical results for tree-based vector quantization guarantee the improvement in average quantization error obtained by partitioning any single region (with a single quantizer) into two disjoints regions (with two quantizers) in the following form (introduced by Freund et al. (2007) [14]):

**Definition 2.1.** *For a set $S \subset \mathbb{R}^\mathbf{D}$, a region $A$ partitioned into two disjoint regions $\{A_l, A_r\}$, and a data-dependent quantity $\beta > 1$, the quantization error improvement is characterized by:*

$$\mathcal{V}_S(\{A_l, A_r\}) < (1 - 1/\beta) \mathcal{V}_S(A). \tag{3}$$

The quantization performance depends inversely on the data-dependent quantity $\beta$ – lower $\beta$ implies better quantization. We present the definition of $\beta$ for different BSP-trees in Table 1. For the *PA*-tree, $\beta$ depends on the ratio of the sum of the eigenvalues of the covariance matrix of data $(A \cap S)$ to the principal eigenvalue. The improvement rate $\beta$ for the *RP*-tree depends on the covariance dimension of the data in the node $A$ ($\beta = O(\mathbf{d}_c)$) [7], which roughly corresponds to the lowest dimensionality of an affine plane that captures most of the data covariance. The *2M*-tree does not have an explicit $\beta$ but it has the optimal theoretical improvement rate for a single partition because the 2-means clustering ob-

| Tree | Definition of $\beta$ |
|------|------------------------|
| *PA*-tree | $O(\varrho^2)$: $\varrho \doteq \left(\sum_{i=1}^\mathbf{D} \lambda_i\right)/\lambda_1$ |
| *RP*-tree | $O(\mathbf{d}_c)$ |
| *kd*-tree | $\times$ |
| *2M*-tree | optimal (smallest possible) |
| *MM*-tree* | $O(\rho)$: $\rho \doteq \left(\sum_{i=1}^\mathbf{D} \lambda_i\right)/\gamma^2$ |

Table 1: $\beta$ **for various trees.** $\lambda_1, \ldots, \lambda_\mathbf{D}$ are the sorted eigenvalues of the covariance matrix of $A \cap S$ in descending order, and $\mathbf{d}_c < \mathbf{D}$ is the covariance dimension of $A \cap S$. The results for *PA*-tree and *2M*-tree are due to Verma et al. (2009) [2]. The *PA*-tree result can be improved to $O(\varrho)$ from $O(\varrho^2)$ with an additional assumption [2]. The *RP*-tree result is in Freund et al. (2007) [14], which also has the precise definition of $\mathbf{d}_c$. We establish the result for *MM*-tree in Section 4. $\gamma$ is the margin size of the large margin partition. No such guarantee for *kd*-trees is known to us.

jective is equal to $|A_l|\mathcal{V}(A_l) + |A_r|\mathcal{V}(A_r)$ and minimizing this objective maximizes $\beta$. The 2-means problem is NP-hard and an approximate solution is used in practice. These theoretical results are valid under the condition that there are no outliers in $A \cap S$. This is characterized as $\max_{x,y \in A \cap S} \|x - y\|^2 \leq \eta \mathcal{V}_S(A)$ for a fixed $\eta > 0$. This notion of the absence of outliers was first introduced for the theoretical analysis of the *RP*-trees [7]. Verma et al. (2009) [2] describe outliers as "points that are much farther away from the mean than the typical distance-from-mean". In this situation, an alternate type of partition is used to remove these outliers that are farther away

from the mean than expected. For $\eta \geq 8$, this alternate partitioning is guaranteed to reduce the data diameter ($\max_{x,y \in A \cap S} \|x - y\|$) of the resulting nodes by a constant fraction [7, Lemma 12], and can be used until a region contain no outliers, at which point, the usual hyperplane partition can be used with their respective theoretical quantization guarantees. The implicit assumption is that the alternate partitioning scheme is employed rarely.

These results for BSP-tree quantization performance indicate that different heuristics are adaptive to different properties of the data. However, no existing theoretical result relates this performance of BSP-trees to their search performance. Making the precise connection between the quantization performance and the search performance of these BSP-trees is a contribution of this paper.

# 3   Approximation guarantees for BSP-tree search

In this section, we formally present the data and tree dependent performance guarantees on the search with BSP-trees using Algorithm 1. The quality of nearest-neighbor search can be quantized in two ways – (i) distance error and (ii) rank of the candidate neighbor. We present guarantees for both notions of search error[2]. For a query $q$ and a set of points $S$ and a neighbor candidate $p \in S$, *distance error* $\epsilon(q) = \frac{\|q-p\|}{\min_{r \in S} \|q-r\|} - 1$, and *rank* $\tau(q) = |\{r \in S : \|q - r\| < \|q - p\|\}| + 1$.

Algorithm 1 requires the query traversal depth $l$ as an input. The search runtime is $O(l + (n/2^l))$. The depth can be chosen based on the desired runtime. Equivalently, the depth can be chosen based on the desired number of candidates $m$; for a balanced binary tree on a dataset $S$ of $n$ points with leaf nodes containing a single point, the appropriate depth $l = \log_2 n - \lceil \log_2 m \rceil$. We will be building on the existing results on vector quantization error [2] to present the worst case error guarantee for Algorithm 1. We need the following definitions to precisely state our results:

**Definition 3.1.** *An $\omega$-balanced split partitioning a region $A$ into disjoint regions $\{A_1, A_2\}$ implies* $||A_1 \cap S| - |A_2 \cap S|| \leq \omega |A \cap S|$.

For a balanced tree corresponding to recursive median splits, such as the *PA*-tree and the *kd*-tree, $\omega \approx 0$. Non-zero values of $\omega \ll 1$, corresponding to approximately balanced trees, allow us to potentially adapt better to some structure in the data at the cost of slightly losing the tree balance. For the *MM*-tree (discussed in detail in Section 4), $\omega$-balanced splits are enforced for any specified value of $\omega$. Approximately balanced trees have a depth bound of $O(\log n)$ [8, Theorem 3.1]. For a tree with $\omega$-balanced splits, the worst case runtime of Algorithm 1 is $O\left(l + \left(\frac{1+\omega}{2}\right)^l n\right)$. For the *2M*-tree, $\omega$-balanced splits are not enforced. Hence the actual value of $\omega$ could be high for a *2M*-tree.

**Definition 3.2.** *Let $\mathcal{B}_{\ell_2}(p, \Delta) = \{r \in S : \|p - r\| < \Delta\}$ denote the points in $S$ contained in a ball of radius $\Delta$ around some $p \in S$ with respect to the $\ell_2$ metric. The **expansion constant** of $(S, \ell_2)$ is defined as the smallest $c \geq 2$ such $\left|\mathcal{B}_{\ell_2}(p, 2\Delta)\right| \leq c \left|\mathcal{B}_{\ell_2}(p, \Delta)\right| \; \forall p \in S$ and $\forall \Delta > 0$.*

Bounded expansion constants correspond to *growth-restricted metrics* [15]. The expansion constant characterizes the data distribution, and $c \sim 2^{O(\mathbf{d})}$ where $\mathbf{d}$ is the *doubling dimension* of the set $S$ with respect to the $\ell_2$ metric. The relationship is exact for points on a $\mathbf{D}$-dimensional grid (i.e., $c = \Theta(2^{\mathbf{D}})$). Equipped with these definitions, we have the following guarantee for Algorithm 1:

**Theorem 3.1.** *Consider a dataset $S \subset \mathbb{R}^{\mathbf{D}}$ of $n$ points with $\psi = \frac{1}{2n^2} \sum_{x,y \in S} \|x - y\|^2$, the BSP tree $T$ built on $S$ and a query $q \in \mathbb{R}^{\mathbf{D}}$ with the following conditions :*

*(C1) Let $(A \cap (S \cup \{q\}), \ell_2)$ have an expansion constant at most $\tilde{c}$ for any convex set $A \subset \mathbb{R}^{\mathbf{D}}$.*

*(C2) Let $T$ be complete till a depth $L < \left(\log_2 \frac{n}{\tilde{c}}\right) / (1 - \log_2(1 - \omega))$ with $\omega$-balanced splits.*

*(C3) Let $\beta^*$ correspond to the worst quantization error improvement rate over all splits in $T$.*

*(C4) For any node $A$ in the tree $T$, let $\max_{x,y \in A \cap S} \|x - y\|^2 \leq \eta \mathcal{V}_S(A)$ for a fixed $\eta \geq 8$.*

*For $\alpha = 1/(1 - \omega)$, the upper bound $d^u$ on the distance of $q$ to the neighbor candidate $p$ returned by Algorithm 1 with depth $l \leq L$ is given by*

$$\|q - p\| \leq d^u = \frac{2\sqrt{\eta\psi} \cdot (2\alpha)^{l/2} \cdot \exp(-l/2\beta^*)}{(n/(2\alpha)^l)^{1/\log_2 \tilde{c}} - 2}. \tag{4}$$

Now $\eta$ is fixed, and $\psi$ is fixed for a dataset $S$. Then, for a fixed $\omega$, this result implies that between two types of BSP-trees on the same set and the same query, Algorithm 1 has a better worst-case guarantee on the candidate-neighbor distance for the tree with better quantization performance (smaller $\beta^*$). Moreover, for a particular tree with $\beta^* \geq \log_2 e$, $d^u$ is non-decreasing in $l$. This is expected because as we traverse down the tree, we can never reduce the candidate neighbor distance. At the root level ($l = 0$), the candidate neighbor is the nearest-neighbor. As we descend down the tree, the candidate neighbor distance will worsen if a tree split separates the query from its closer neighbors. This behavior is implied in Equation (4). For a chosen depth $l$ in Algorithm 1, the candidate neighbor distance is inversely proportional to $\left(n/(2\alpha)^l\right)^{1/\log_2 \tilde{c}}$, implying deteriorating bounds $d^u$ with increasing $\tilde{c}$. Since $\log_2 \tilde{c} \sim O(\mathbf{d})$, larger intrinsic dimensionality implies worse guarantees as expected from the curse of dimensionality. To prove Theorem 3.1, we use the following result:

**Lemma 3.1.** *Under the conditions of Theorem 3.1, for any node $A$ at a depth $l$ in the BSP-tree $T$ on $S$, $\mathcal{V}_S(A) \leq \psi \left(2/(1-\omega)\right)^l \exp(-l/\beta^*)$.*

This result is obtained by recursively applying the quantization error improvement in Definition 2.1 over $l$ levels of the tree (the proof is in Appendix A).

*Proof of Theorem 3.1.* Consider the node $A$ at depth $l$ in the tree containing $q$, and let $m = |A \cap S|$. Let $D = \max_{x,y \in A \cap S} \|x - y\|$, let $d = \min_{x \in A \cap S} \|q - x\|$, and let $\mathcal{B}_{\ell_2}(q, \Delta) = \{x \in A \cap (S \cup \{q\}) \colon \|q - x\| < \Delta\}$. Then, by the Definition 3.2 and condition *C1*,

$$\left|\mathcal{B}_{\ell_2}(q, D + d)\right| \leq \tilde{c}^{\log_2 \left\lceil \frac{D+d}{d} \right\rceil} |\mathcal{B}_{\ell_2}(q, d)| = \tilde{c}^{\log_2 \left\lceil \frac{D+d}{d} \right\rceil} \leq \tilde{c}^{\log_2 \left(\frac{D+2d}{d}\right)},$$

where the equality follows from the fact that $\mathcal{B}_{\ell_2}(q, d) = \{q\}$. Now $\left|\mathcal{B}_{\ell_2}(q, D + d)\right| \geq m$. Using this above gives us $m^{1/\log_2 \tilde{c}} \leq (D/d) + 2$. By condition *C2*, $m^{1/\log_2 \tilde{c}} > 2$. Hence we have $d \leq D/(m^{1/\log_2 \tilde{c}} - 2)$. By construction and condition *C4*, $D \leq \sqrt{\eta \mathcal{V}_S(A)}$. Now $m \geq n/(2\alpha)^l$. Plugging this above and utilizing Lemma 3.1 gives us the statement of Theorem 3.1. ☐

**Nearest-neighbor search error guarantees.** Equipped with the bound on the candidate-neighbor distance, we bound the worst-case nearest-neighbor search errors as follows:

**Corollary 3.1.** *Under the conditions of Theorem 3.1, for any query $q$ at a desired depth $l \leq L$ in Algorithm 1, the **distance error** $\epsilon(q)$ is bounded as $\epsilon(q) \leq (d^u/d_q^*) - 1$, and the **rank** $\tau(q)$ is bounded as $\tau(q) \leq \tilde{c}^{\left\lceil \log_2(d^u/d_q^*) \right\rceil}$, where $d_q^* = \min_{r \in S} \|q - r\|$.*

*Proof.* The distance error bound follows from the definition of distance error. Let $R = \{r \in S \colon \|q - r\| < d^u\}$. By definition, $\tau(q) \leq |R| + 1$. Let $\mathcal{B}_{\ell_2}(q, \Delta) = \{x \in (S \cup \{q\}) \colon \|q - x\| < \Delta\}$. Since $\mathcal{B}_{\ell_2}(q, d^u)$ contains $q$ and $R$, and $q \notin S$, $|\mathcal{B}_{\ell_2}(q, d^u)| = |R| + 1 \geq \tau(q)$. From Definition 3.2 and Condition *C1*, $|\mathcal{B}_{\ell_2}(q, d^u)| \leq \tilde{c}^{\left\lceil \log_2(d^u/d_q^*) \right\rceil} |\mathcal{B}_{\ell_2}(q, d_q^*)|$. Using the fact that $|\mathcal{B}_{\ell_2}(q, d_q^*)| = |\{q\}| = 1$ gives us the upper bound on $\tau(q)$. ☐

The upper bounds on both forms of search error are directly proportional to $d^u$. Hence, the BSP-tree with better quantization performance has better search performance guarantees, and increasing traversal depth $l$ implies less computation but worse performance guarantees. Any dependence of this approximation guarantee on the ambient data dimensionality is subsumed by the dependence on $\beta^*$ and $\tilde{c}$. While our result bounds the worst-case performance of Algorithm 1, an average case performance guarantee on the distance error is given by $\mathbb{E}_q \epsilon(q) \leq d^u \mathbb{E}_q \left(1/d_q^*\right) - 1$, and on the rank is given by $\mathbb{E}_q \tau(q) \leq \tilde{c}^{\lceil \log_2 d^u \rceil} \left(\mathbb{E}_q c^{-\left(\log_2 d_q^*\right)}\right)$, since the expectation is over the queries $q$ and $d^u$ does not depend on $q$. For the purposes of relative comparison among BSP-trees, the bounds on the expected error depend solely on $d^u$ since the term within the expectation over $q$ is tree independent.

**Dependence of the nearest-neighbor search error on the partition margins.** The search error bounds in Corollary 3.1 depend on the true nearest-neighbor distance $d_q^*$ of any query $q$ of which we have no prior knowledge. However, if we partition the data with a large margin split, then we can say that either the candidate neighbor is the true nearest-neighbor of $q$ or that $d_q^*$ is greater than the size of the margin. We characterize the influence of the margin size with the following result:

**Corollary 3.2.** *Consider the conditions of Theorem 3.1 and a query $q$ at a depth $l \leq L$ in Algorithm 1. Further assume that $\gamma$ is the smallest margin size on both sides of any partition in the tree $T$. Then the distance error is bounded as $\epsilon(q) \leq d^u/\gamma - 1$, and the rank is bounded as $\tau(q) \leq \tilde{c}^{\lceil \log_2(d^u/\gamma) \rceil}$.*

This result indicates that if the split margins in a BSP-tree can be increased without adversely affecting its quantization performance, the BSP-tree will have improved nearest-neighbor error guarantees

for the Algorithm 1. This motivated us to consider the *max-margin tree* [8], a BSP-tree that explicitly maximizes the margin of the split for every split in the tree.

**Explanation of the conditions in Theorem 3.1.** Condition *C1* implies that for *any* convex set $A \subset \mathbb{R}^{\mathbf{D}}$, $((A \cap (S \cup \{q\})), \ell_2)$ has an expansion constant at most $\tilde{c}$. A bounded $\tilde{c}$ implies that no subset of $(S \cup \{q\})$, contained in a convex set, has a very high expansion constant. This condition implies that $((S \cup \{q\}), \ell_2)$ also has an expansion constant at most $\tilde{c}$ (since $(S \cup \{q\})$ is contained in its convex hull). However, if $(S \cup \{q\}, \ell_2)$ has an expansion constant $c$, this does not imply that the data lying within any convex set has an expansion constant at most $c$. Hence a bounded expansion constant assumption for $(A \cap (S \cup \{q\}), \ell_2)$ for every convex set $A \subset \mathbb{R}^{\mathbf{D}}$ is stronger than a bounded expansion constant assumption for $(S \cup \{q\}, \ell_2)$[3]. Condition *C2* ensures that the tree is complete so that for every query $q$ and a depth $l \leq L$, there exists a large enough tree node which contains $q$. Condition *C3* gives us the worst quantization error improvement rate over all the splits in the tree.

Condition *C4* implies that the squared data diameter of any node $A$ ($\max_{x,y \in A \cap S} \|x - y\|^2$) is within a constant factor of its quantization error $\mathcal{V}_S(A)$. This refers to the assumption that the node $A$ contains no outliers as described in Section 3 and only hyperplane partitions are used and their respective quantization improvement guarantees presented in Section 2 (Table 1) hold. By placing condition *C4*, we ignore the alternate partitioning scheme used to remove outliers for simplicity of analysis. If we allow a small fraction of the partitions in the tree to be this alternate split, a similar result can be obtained since the alternate split is the same for all BSP-tree. For two different kinds of hyperplane splits, if alternate split is invoked the same number of times in the tree, the difference in their worst-case guarantees for both the trees would again be governed by their worst-case quantization performance ($\beta^*$). However, for any fixed $\eta$, a harder question is whether one type of hyperplane partition violates the inlier condition more often than another type of partition, resulting in more alternate partitions. And we do not yet have a theoretical answer for this[4].

**Empirical validation.** We examine our theoretical results with 4 datasets – OPTDIGITS ($\mathbf{D} = 64$, $n = 3823$, 1797 queries), TINY IMAGES ($\mathbf{D} = 384$, $n = 5000$, 1000 queries), MNIST ($\mathbf{D} = 784$, $n = 6000$, 1000 queries), IMAGES ($\mathbf{D} = 4096$, $n = 500$, 150 queries). We consider the following BSP-trees: *kd*-tree, random-projection (*RP*) tree, principal axis (*PA*) tree, two-means (*2M*) tree and max-margin (*MM*) tree. We only use hyperplane partitions for the tree construction. This is because, firstly, the check for the presence of outliers ($\Delta_S^2(A) > \eta \mathcal{V}_S(A)$) can be computationally expensive for large $n$, and, secondly, the alternate partition is mostly for the purposes of obtaining theoretical guarantees. The implementation details for the different tree constructions are presented in Appendix C. The performance of these BSP-trees are presented in Figure 2. Trees with missing data points for higher depth levels (for example, *kd*-tree in Figure 2(a) and *2M*-tree in Figures 2 (b) & (c)) imply that we were unable to grow complete BSP-trees beyond that depth.

The quantization performance of the *2M*-tree, *PA*-tree and *MM*-tree are significantly better than the performance of the *kd*-tree and *RP*-tree and, as suggested by Corollary 3.1, this is also reflected in their search performance. The *MM*-tree has comparable quantization performance to the *2M*-tree and *PA*-tree. However, in the case of search, the *MM*-tree outperforms *PA*-tree in all datasets. This can be attributed to the large margin partitions in the *MM*-tree. The comparison to *2M*-tree is not as apparent. The *MM*-tree and *PA*-tree have $\omega$-balanced splits for small $\omega$ enforced algorithmically, resulting in bounded depth and bounded computation of $O(l + n(1 + \omega)^l/2^l)$ for any given depth $l$. No such balance constraint is enforced in the 2-means algorithm, and hence, the *2M*-tree can be heavily unbalanced. The absence of complete BSP *2M*-tree beyond depth 4 and 6 in Figures 2 (b) & (c) respectively is evidence of the lack of balance in the *2M*-tree. This implies possibly more computation and hence lower errors. Under these conditions, the *MM*-tree with an explicit balance constraint performs comparably to the *2M*-tree (slightly outperforming in 3 of the 4 cases) while still maintaining a balanced tree (and hence returning smaller candidate sets on average).

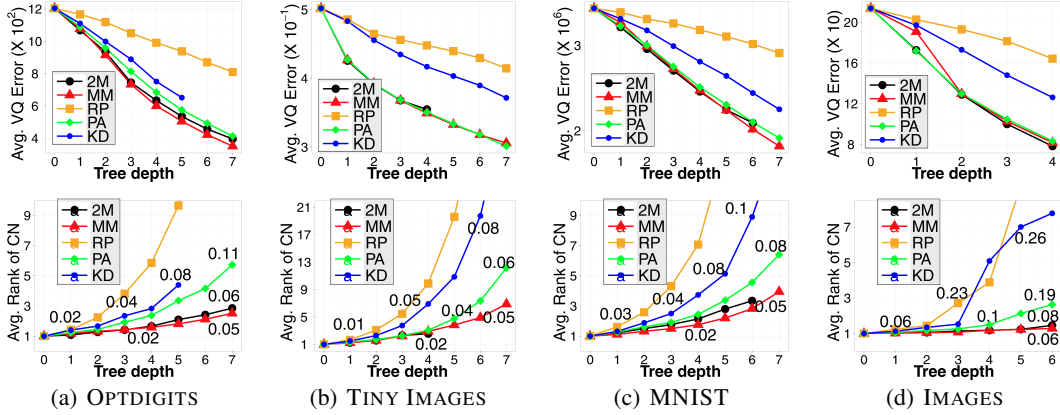

(a) OPTDIGITS&emsp;&emsp;(b) TINY IMAGES&emsp;&emsp;(c) MNIST&emsp;&emsp;(d) IMAGES

Figure 2: **Performance of BSP-trees with increasing traversal depth**. The top row corresponds to quantization performance of existing trees and the bottom row presents the nearest-neighbor error (in terms of mean rank $\tau$ of the candidate neighbors (CN)) of Algorithm 1 with these trees. The nearest-neighbor search error graphs are also annotated with the mean distance-error of the CN (*please view in color*).

## 4&emsp;Large margin BSP-tree

We established that the search error depends on the quantization performance and the partition margins of the tree. The *MM*-tree explicitly maximizes the margin of every partition and empirical results indicate that it has comparable performance to the *2M*-tree and *PA*-tree in terms of the quantization performance. In this section, we establish a theoretical guarantee for the *MM*-tree quantization performance. The large margin split in the *MM*-tree is obtained by performing max-margin clustering (MMC) with 2 clusters. The task of MMC is to find the optimal hyperplane $(\mathbf{w}^*, b^*)$ from the following optimization problem[5] given a set of points $S = \{x_1, x_2, \ldots, x_m\} \subset \mathbb{R}^{\mathbf{D}}$:

$$\min_{\mathbf{w}, b, \xi_i} \frac{1}{2} \|\mathbf{w}\|_2^2 + C \sum_{i=1}^{m} \xi_i \tag{5}$$

$$s.t. \qquad |\langle \mathbf{w}, x_i \rangle + b| \geq 1 - \xi_i, \ \ \xi_i \geq 0 \ \ \forall i = 1, \ldots, m \tag{6}$$

$$-\omega m \leq \sum_{i=1}^{m} \text{sgn}(\langle \mathbf{w}, x_i \rangle + b) \leq \omega m. \tag{7}$$

MMC finds a soft max-margin split in the data to obtain two clusters separated by a large (soft) margin. The balance constraint (Equation (7)) avoids trivial solutions and enforces an $\omega$-balanced split. The margin constraints (Equation (6)) enforce a robust separation of the data. Given a solution to the MMC, we establish the following quantization error improvement rate for the *MM*-tree:

**Theorem 4.1.** *Given a set of points $S \subset \mathbb{R}^{\mathbf{D}}$ and a region $A$ containing $m$ points, consider an $\omega$-balanced max-margin split $(\mathbf{w}, b)$ of the region $A$ into $\{A_l, A_r\}$ with at most $\alpha m$ support vectors and a split margin of size $\gamma = 1/\|\mathbf{w}\|$. Then the quantization error improvement is given by:*

$$\mathcal{V}_S(\{A_l, A_r\}) \leq \left(1 - \frac{\gamma^2 (1 - \alpha)^2 \left(\frac{1-\omega}{1+\omega}\right)}{\sum_{i=1}^{\mathbf{D}} \lambda_i}\right) \mathcal{V}_S(A), \tag{8}$$

*where $\lambda_1, \ldots, \lambda_{\mathbf{D}}$ are the eigenvalues of the covariance matrix of $A \cap S$.*

The result indicates that larger margin sizes (large $\gamma$ values) and a smaller number of support vectors (small $\alpha$) implies better quantization performance. Larger $\omega$ implies smaller improvement, but $\omega$ is generally restricted algorithmically in MMC. If $\gamma = O(\sqrt{\lambda_1})$ then this rate matches the best possible quantization performance of the *PA*-tree (Table 1). We do assume that we have a feasible solution to the MMC problem to prove this result. We use the following result to prove Theorem 4.1:

**Proposition 4.1.** *[7, Lemma 15] Give a set $S$, for any partition $\{A_1, A_2\}$ of a set $A$,*

$$\mathcal{V}_S(A) - \mathcal{V}_S(\{A_1, A_2\}) = \frac{|A_1 \cap S||A_2 \cap S|}{|A \cap S|^2} \|\mu(A_1) - \mu(A_2)\|^2, \tag{9}$$

*where $\mu(A)$ is the centroid of the points in the region $A$.*

This result [7] implies that the improvement in the quantization error depends on the distance between the centroids of the two regions in the partition.

*Proof of Theorem 4.1.* For a feasible solution $(\mathbf{w}, b, \xi_i|_{i=1,\dots,m})$ to the MMC problem,

$$\sum_{i=1}^{m} |\langle \mathbf{w}, x_i \rangle + b| \geq m - \sum_{i=1}^{m} \xi_i.$$

Let $\tilde{x}_i = \langle \mathbf{w}, x_i \rangle + b$ and $m_p = |\{i \colon \tilde{x}_i > 0\}|$ and $m_n = |\{i \colon \tilde{x}_i \leq 0\}|$ and $\tilde{\mu}_p = (\sum_{i \colon \tilde{x}_i > 0} \tilde{x}_i)/m_p$ and $\tilde{\mu}_n = (\sum_{i \colon \tilde{x}_i \leq 0} \tilde{x}_i)/m_n$. Then $m_p \tilde{\mu}_p - m_n \tilde{\mu}_n \geq m - \sum_i \xi_i$.

Without loss of generality, we assume that $m_p \geq m_n$. Then the balance constraint (Equation (7)) tells us that $m_p \leq m(1+\omega)/2$ and $m_n \geq m(1-\omega)/2$. Then $\tilde{\mu}_p - \tilde{\mu}_n + \omega(\tilde{\mu}_p + \tilde{\mu}_n) \geq 2 - \frac{2}{m}\sum_i \xi_i$.

Since $\tilde{\mu}_p > 0$ and $\mu_n \leq 0$, $|\tilde{\mu}_p + \tilde{\mu}_n| \leq (\tilde{\mu}_p - \tilde{\mu}_n)$. Hence $(1 + \omega)(\tilde{\mu}_p - \tilde{\mu}_n) \geq 2 - \frac{2}{m}\sum_i \xi_i$. For an unsupervised split, the data is always separable since there is no misclassification. This implies that $\xi_i^* \leq 1 \forall i$. Hence,

$$\tilde{\mu}_p - \tilde{\mu}_n \geq \left(2 - \frac{2}{m}|\{i \colon \xi_i > 0\}|\right)/(1+\omega) \geq 2\left(\frac{1-\alpha}{1+\omega}\right), \tag{10}$$

since the term $|\{i \colon \xi_i > 0\}|$ corresponds to the number of support vectors in the solution.

Cauchy-Schwartz implies that $\|\mu(A_l) - \mu(A_r)\| \geq |\langle \mathbf{w}, \mu(A_l) - \mu(A_r) \rangle|/\|\mathbf{w}\| = (\tilde{\mu}_p - \tilde{\mu}_n)\gamma$, since $\tilde{\mu}_n = \langle \mathbf{w}, \mu(A_l) \rangle + b$ and $\tilde{\mu}_p = \langle \mathbf{w}, \mu(A_r) \rangle + b$. From Equation (10), we can say that $\|\mu(A_l) - \mu(A_r)\|^2 \geq 4\gamma^2 (1-\alpha)^2 / (1+\omega)^2$. Also, for $\omega$-balanced splits, $|A_l||A_r| \geq (1-\omega^2)m^2/4$. Combining these into Equation (9) from Proposition 4.1, we have

$$\mathcal{V}_S(A) - \mathcal{V}_S(\{A_l, A_r\}) \geq (1-\omega^2)\gamma^2 \left(\frac{1-\alpha}{1+\omega}\right)^2 = \gamma^2 (1-\alpha)^2 \left(\frac{1-\omega}{1+\omega}\right). \tag{11}$$

Let $\mathrm{Cov}(A \cap S)$ be the covariance matrix of the data contained in region $A$ and $\lambda_1, \dots, \lambda_\mathbf{D}$ be the eigenvalues of $\mathrm{Cov}(A \cap S)$. Then, we have:

$$\mathcal{V}_S(A) = \frac{1}{|A \cap S|} \sum_{x \in A \cap S} \|x - \mu(A)\|^2 = \mathrm{tr}\left(\mathrm{Cov}(A \cap S)\right) = \sum_{i=1}^{\mathbf{D}} \lambda_i.$$

Then dividing Equation (11) by $\mathcal{V}_S(A)$ gives us the statement of the theorem. $\qquad\square$

## 5  Conclusions and future directions

Our results theoretically verify that BSP-trees with better vector quantization performance and large partition margins do have better search performance guarantees as one would expect. This means that the best BSP-tree for search on a given dataset is the one with the best combination of good quantization performance (low $\beta^*$ in Corollary 3.1) and large partition margins (large $\gamma$ in Corollary 3.2). The *MM*-tree and the *2M*-tree appear to have the best empirical performance in terms of the search error. This is because the *2M*-tree explicitly minimizes $\beta^*$ while the *MM*-tree explicitly maximizes $\gamma$ (which also implies smaller $\beta^*$ by Theorem 4.1). Unlike the *2M*-tree, the *MM*-tree explicitly maintains an approximately balanced tree for better worst-case search time guarantees.

However, the general dimensional large margin partitions in the *MM*-tree construction can be quite expensive. But the idea of large margin partitions can be used to enhance any simpler space partition heuristic – for any chosen direction (such as along a coordinate axis or along the principal eigenvector of the data covariance matrix), a one dimensional large margin split of the projections of the points along the chosen direction can be obtained very efficiently for improved search performance.

This analysis of search could be useful beyond BSP-trees. Various heuristics have been developed to improve locality-sensitive hashing (LSH) [10]. The plain-vanilla LSH uses random linear projections and random thresholds for the hash-table construction. The data can instead be projected along the top few eigenvectors of the data covariance matrix. This was (empirically) improved upon by learning an orthogonal rotation of the projected data to minimize the quantization error of each bin in the hash-table [17]. A nonlinear hash function can be learned using a restricted Boltzmann machine [18]. If the similarity graph of the data is based on the Euclidean distance, spectral hashing [19] uses a subset of the eigenvectors of the similarity graph Laplacian. Semi-supervised hashing [20] incorporates given pairwise semantic similarity and dissimilarity constraints. The structural SVM framework has also been used to learn hash functions [21]. Similar to the choice of an appropriate BSP-tree for search, the best hashing scheme for any given dataset can be chosen by considering the quantization performance of the hash functions and the margins between the bins in the hash tables. We plan to explore this intuition theoretically and empirically for LSH based search schemes.

## Footnotes

[1] This intuitive connection is widely believed but never rigorously established to the best of our knowledge.

[2]The distance error corresponds to the relative error in terms of the actual distance values. The rank is one more than the number of points in $S$ which are better neighbor candidates than $p$. The nearest-neighbor of $q$ has rank 1 and distance error 0. The appropriate notion of error depends on the search application.

[3]A subset of a growth-restricted metric space $(S, \ell_2)$ may not be growth-restricted. However, in our case, we are not considering all subsets; we only consider subsets of the form $(A \cap S)$ where $A \subset \mathbb{R}^{\mathbf{D}}$ is a convex set. So our condition does not imply that all subsets of $(S, \ell_2)$ are growth-restricted.

[4]We empirically explore the effect of the tree type on the violation of the inlier condition (*C4*) in Appendix B. The results imply that for any fixed value of $\eta$, almost the same number of alternate splits would be invoked for the construction of different types of trees on the same dataset. Moreover, with $\eta \geq 8$, for only one of the datasets would a significant fraction of the partitions in the tree (of any type) need to be the alternate partition.

[5]This is an equivalent formulation [16] to the original form of max-margin clustering proposed by Xu et al. (2005) [9]. The original formulation also contains the labels $y_i$s and optimizes over it. We consider this form of the problem since it makes our analysis easier to follow.

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
