[Supplementary Material]

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

# A    Proof of Lemma 3.1

We need the following result to prove Lemma 3.1.

**Fact A.1.** *For any set of points $S$ and a region $A$ containing $m$ of the points in $S$ and quantized by the centroid $\mu(A)$ of these points,*

$$\mathcal{V}_S(A) = \frac{1}{2}\frac{1}{m^2} \sum_{x,y \in A \cap S} \|x - y\|^2. \tag{12}$$

*Proof.*

$$
\begin{aligned}
\frac{1}{m^2} \sum_{x,y \in (A \cap S)} \|x - y\|^2 &= \frac{1}{m^2} \sum_{x,y \in (A \cap S)} \|x - \mu(A) + \mu(A) - y\|^2 \\
&= \frac{1}{m^2} \sum_{x,y \in (A \cap S)} \|x - \mu(A)\|^2 + \frac{1}{m^2} \sum_{x,y \in (A \cap S)} \|y - \mu(A)\|^2 \\
&\quad + \frac{2}{m^2} \sum_{x,y \in (A \cap S)} \langle x - \mu(A), \mu(A) - y \rangle \\
&= \frac{2}{m} \sum_{x \in (A \cap S)} \|x - \mu(A)\|^2 + \frac{2}{m^2} \sum_{x,y \in (A \cap S)} \langle x - \mu(A), \mu(A) - y \rangle \\
&\quad \text{since} \sum_{x,y \in (A \cap S)} \|x - \mu(A)\|^2 = m \sum_{x \in (A \cap S)} \|x - \mu(A)\|^2 \\
&= \frac{2}{m} \sum_{x \in (A \cap S)} \|x - \mu(A)\|^2 + \frac{2}{m^2} \sum_{y \in (A \cap S)} \langle \sum_{x \in (A \cap S)} (x - \mu(A)), \mu(A) - y \rangle \\
&= \frac{2}{m} \sum_{x \in (A \cap S)} \|x - \mu(A)\|^2 \quad \text{since} \sum_{x \in (A \cap S)} (x - \mu(A)) = \mathbf{0} \\
&= 2\mathcal{V}_S(A).
\end{aligned}
$$

$\square$

We state Lemma 3.1 again here:

**Lemma A.1.** *Under the conditions of Theorem 3.1, for any node $A$ at a depth $l$ in the BSP-tree $T$ on $S$,*

$$\mathcal{V}_S(A) \leq \psi \left( \frac{2}{1 - \omega} \right)^l \exp(-l/\beta^*). \tag{13}$$

*Proof.* By definition, for any node $A$ at depth $l$, its sibling node $A_s$ and parent node $A_p$,

$$\mathcal{V}_S(\{A, A_s\}) \leq \left( 1 - \frac{1}{\beta^*} \right) \mathcal{V}_S(A_p).$$

For $\omega$-balanced splits, assuming that $|A| < |A_s|$,

$$\mathcal{V}_S(\{A, A_s\}) = \frac{1 - \omega}{2} \mathcal{V}_S(A) + \frac{1 + \omega}{2} \mathcal{V}_S(A_s) \leq \left( 1 - \frac{1}{\beta^*} \right) \mathcal{V}_S(A_p).$$

This gives us the following:

$$\frac{\mathcal{V}_S(A)}{\mathcal{V}_S(A_p)} \leq \left( \frac{2}{1 - \omega} \right) \frac{\mathcal{V}_S(\{A, A_s\})}{\mathcal{V}_S(A_p)} \leq \left( \frac{2}{1 - \omega} \right) \left( 1 - \frac{1}{\beta^*} \right).$$

The same holds for $A_s$ and for every node at depth $l$ and its corresponding parent at depth $l - 1$. Recursing this up to the root node of the tree $A_0$, we have:

$$\frac{\mathcal{V}_S(A)}{\mathcal{V}_S(A_0)} \leq \left( \frac{2}{1 - \omega} \right)^l \left( 1 - \frac{1}{\beta^*} \right)^l \leq \left( \frac{2}{1 - \omega} \right)^l e^{-l/\beta^*}. \tag{14}$$

for any node $A$ at a depth $l$ of the tree. Using Fact A.1, $\mathcal{V}_S(A_0) = \psi$. Substituting this in Equation (14) gives us the statement of Lemma A.1. $\square$

# B  Empirical values of $\Delta_S^2(A)/\mathcal{V}_S(A)$

We consider the $4$ datasets used for evaluation and compute the ratio $\Delta_S^2(A)/\mathcal{V}_S(A)$ for every node $A$ in the tree. We consider the *2M*-tree, the *PA*-tree and the *kd*-tree. Figure 3 present the actual values of this ratio. The top row presents every recorded value of this ratio while the bottom row averages this value for every node at the same level/depth of the tree.

Figure 3: **Ratio** $\Delta_S^2(A)/\mathcal{V}_S(A)$. The top row presents the actual ratio values for nodes at different depths in the tree as a scatter plot, and the bottom row presents ratio averaged over all nodes at the same depth (*please view in color*).

The empirical results indicate that in $3$ out of the $4$ datasets (OPTDIGITS, MNIST & IMAGES), the average ratio is not affected by the depth of the tree (as seen by the bottom row). The depth of the tree does increase the spread of $(\Delta_S^2(A)/\mathcal{V}_S(A))$. Only in the TINY IMAGES set does the ratio decrease with increasing depth of the tree.

The thing of notice is that the value of the ratio for the tree nodes do not differ significantly between the trees; *2M*-tree, *PA*-tree and *kd*-tree appear to have the same values in most of the cases. These results imply that for a chosen (fixed) value of $\eta$ in Theorem 3.1, (almost) the same number of 'split by distance' will be used for the construction of different types of trees on a single dataset. The values of the ratio does vary between datasets.

# C  Implementation details

Lloyd's algorithm [22] is used for approximate 2-means clustering to build the *2M*-tree. *RP*-tree is constructed using the algorithm presented in Fruend et al., 2007 [14, Procedure CHOOSERULE(S), page 5]. We consider 20 random directions for each split and choose the direction with the maximum improvement in quantization error. For *2M*-tree and *RP*-tree, we perform 15 tree constructions and present the results for the tree with the best nearest-neighbor performance. For the *PA*-tree, we use a median split along the principal axis of the covariance of the

| Dataset | $C$ |
|---|---|
| OPTDIGITS | $10^{-3}$ |
| TINY IMAGES | $0.5$ |
| MNIST | $10^{-6}$ |
| IMAGES | $0.05$ |

Table 2: Chosen regularization parameter for *MM*-tree construction.

data, and for the *kd*-tree, we use a median split along the coordinate axis with the highest variance. We use an iterative algorithm to perform the unsupervised max-margin split in the *MM*-tree construction. The algorithm starts with a label assignment of the data, and then iteratively finds a large margin split and updates the labels until convergence. For *MM*-tree, we perform 5 tree constructions and present the results for the tree with the best nearest-neighbor performance. The regularization parameter $C$ is chosen by a grid search along the log scale. The best values for different datasets are presented in Table 2. The same regularization parameter is used for all splits in the tree. The balance constraint $\omega$ is chosen as $0.1$.