[Reviews · NeurIPS 2013]

Submitted by Assigned_Reviewer_6

The paper presents bounds on the search performance of a simple,
tree-based nearest neighbor search algorithm. The bounds depend on the
vector quantization performance on the tree. It is argued that this
result implies that trees with good vector quantization performance
are advantageous for nearest neighbor search. The statement is extended
to large margin splits.

The title of the paper asks "which space partitioning tree to use for
search"? It should better ask "which tree results in the strongest
performance guarantees"? The paper says almost nothing about practical
performance. This is mostly due to the choice of an artificially
simplified search procedure. More often than not, a better guarantee is
an artifact of a certain flavor of analysis or a proof technique, since
we are only talking about upper bounds. If the bounds are not tight
then the bounds say little about which tree to "use" (in practice!).
This paper makes the common mistake of confusing a better performance
guarantee with a guarantee for better performance. This happens at
several spots, e.g., first sentence of section 4.

Algorithm 1 is analyzed in depth. However, I am unsure how relevant
this algorithm is. It descends the tree without backtracking. At the
target depth l it performs exhaustive search. Although this is not
taken into account in the analysis, the final search can be performed
with efficient exact tree search, so, this time with backtracking.

This algorithm does not find the exact nearest neighbor. The obvious
alternative is to apply a heuristic for skipping some of the branches
on the fly. The decisive difference is that in Algorithm 1 the decision
which branches to traverse is not made in a data dependent manner, but
instead based on a pre-specified parameter. This is why personally
I would never use this algorithm. Since all results are restricted to
this algorithm I question the relevance of this paper.

I see that analyzing the computational complexity of a method with
backtracking is surely much harder. I argue that doing so would be a
prerequisite for understanding the behavior of realistic algorithms.
I cannot get rid of the impression that this analysis was conducted
simply because it is possible, and not because it is relevant.

The logic of the conclusion is as follows:
Algorithm 1: good VQ performance => good search performance.
Now Algorithm 1 is not a good search algorithm by itself. When using
more elaborate tree search procedures there remains little to nothing
to conclude from the present analysis. However, the title as well as
other statements in the paper (e.g., top of page 5) indicate that the
conclusion is rather general. I want to emphasize that this is not the
case, and thus this paper does NOT answer the question which search
tree to use for search in practice.

I would appreciate the result if it could help the analysis of more
realistic search procedures. However, I am not an expert on the
analysis of tree search and thus I cannot judge the present paper from
this perspective. Also, this paper does not claim to provide new
methods for analysis, it is all about the theorems. And this makes me
question its value.

The empirical evaluation is weak. Only four data sets are used, and
they are even non-standard. E.g., why is MNIST sub-sampled to 6000
training and 1000 test samples? This is arbitrary, an no reason is
given. This does not help my trust in the evaluation. With low numbers
of training points there is no real need for tree search at all.
I see that the empirical results are nicely in line with the analysis.
However, how about computation time? E.g., a single decision in a
kd-tree is cheaper than in a PA-tree by a factor of O(dim(X)). The
added computation time could be used for backtracking, which could
well give the kd-tree an advantage. So once more, this analysis says
nothing about which tree to use for search with a better algorithm.
Summary: I don't have trust that this simplified analysis will actually answer
the question posed in the title for practical purposes. This is because
a too much simplified search algorithm is considered. This reduces the
relevance of the analysis to nearly zero.

I have just read the author feedback. I find my points addressed
very well and convincingly. I have changed my decision accordingly.
Thanks for the good feedback!

Submitted by Assigned_Reviewer_7

Paper Summary:
The authors formalize and prove the long standing assumption that partition trees with better quantization rates also have better nearest neighbor search performance. Inspired by this, they propose the ‘max margin’ partition tree and relate its search performance with the margin. Experiments with some real world data validate that ‘max margin’ and partition trees with good quantization errors yield good nearest neighbors.

Review:
I really like that the authors were able to formalize the long standing assumption that partition trees with better quantization rates tend to have better nearest neighbor search performance. The formalism and conditions proposed in Theorem 3.1 are intuitive; and the performance is nicely related to the the ‘expansion constant’ (a formalism of intrinsic dimension of the input sample). Although the bound provided in Eq. 4 can be made tighter (see below).


I do have a few suggestions that authors should consider adding to the current text:

- while using the expansion dimension is intuitive, this results in making a strong assumption like condition C1 to hold over _every_ convex set in the underlying space. I believe (I haven't really checked it) that if the authors consider doubling dimension instead, they dont have to explicitly assume C1 to hold, thus improving the overall readability of the text. (perhaps make a note that bounded doubling dimension, also implies the result?)

- perhaps the authors want to add a short discussion on what happens if the input S comes from an underlying distribution.

- why the notation ‘\tilde c’ for expansion constant? consider changing it to ‘c’. (Theorem 3.1)

- I believe that the bound provided in Eq. 4 can be made tighter. Shouldn't the ratio in lines 531--532 always less than 1? This can significantly improve Eq. 14.


Quality:
The paper systematically explores the connection between vector quantization and nearest neighbor search, providing both a sound theory and convincing experiments. The bound seems to be looser than expected.

Clarity:
The paper is written clearly.

Originality:
The work is original.

Significance:
I believe this work is very significant as it formalizes the long standing belief that partition trees with better quantization rates tend to have better nearest neighbor search performance. This work can encourage formal analysis of other fast nearest neighbor schemes.


Update after the author response:
Thanks for the clarifications. In the past week, I came across the following COLT2013 paper that is highly relevant to this submission: "Randomized Partition Trees for Exact Nearest Neighbor Search". The authors should update their discussion and compare their results with this paper as well.
Summary: Authors have done a good job in formalizing the long standing intuition that partition trees with better quantization rates tend to have better nearest neighbor search performance. The authors can do a tighter analysis to improve the bound. Experiments with real data corroborate that the theoretical intuition also works well in practice.

Submitted by Assigned_Reviewer_8

The authors study space partioning trees with the nearest-neighbor search. More
specifically, they study the error caused by dfs search (without backtracking)
and show connection between the error and quantization error of the tree.

The authors work is interesting, however, there seems to be some weaknesses.
Authors demonstate the connection between the search error and the quantization
error by bounding the search error with a (rather complex) bound that has
parameters derived from the quantization error. However, this connection
is implied only if the bound is tight, which doesn't seem to be the case:

Consider the expansion coefficient c in in Theorem 3.1. This coefficient
depends on q. If we select q to be far enough from S: let's say that that
min_{x \in S} |x - p| is bigger than the distance of any two points in S. This
implies that expansion coefficient is n, which forces L in C2 to be 0, making
Theorem 3.1. unusable for this case. The problem with this one is that Theorem
3.1. analysies only upstairs of the distance error. I think a more solid result
could be obtained is the downstairs term is also taken into account
simultaneosly.

The fact that c depends on q implies that Theorem 3.1 cannot be used in
practice to obtain the actual error bounds while doing the actual search since
computing expansion coefficient seems to be very computationally expensive.
Theorem 3.1. can be only used as a theoretical evidence for the connection
between the search error and the quantization error.

Let's make an assumption now that q is actually "within" S, that the expansion
coefficient doesn't change that much when we add q into S. Consider the
following case: copy S and transpose the copy from S such that they both copies
are fully separated. The expansion coefficient should stay about the same, as
well as \omega (since any reasonable tree would first separate the copies from
each other, giving \omega = 0, for the first level). While the quantization
improvement rate will be excellent on the first level, it will be as bad on the
next levels as with original S. Consequently, \beta \beta will stay the same.
As we move the copies from each other away. \psi will get larger, and the bound
will get loose. I think a better approach would not to consider \psi and \beta
globally but instead try to work on a "local level".

While these cases may not happen in practice, I would like to see authors
demonstrating how tight is the bound, for example, empirically, similar to
Figure 2.

Techical writing could be improved. The authors tend to use accents and
indices in cases where they are not really needed. For example,

\tilde{c} -> c

B_{l_2} -> B

use less exotic caligraphy for intrinsic dimenstion d and full dimension D

027: - -> ---

Definition 2.1. is a bit messy. It would be better to just say that
quantization error improvement rate is equal to
beta = V_S(A) / (V_S(A) - V_S(A_l, A_r))

C2: what do you mean by complete?
C4: in Theorem 3.1. are all nodes in T included or only leaves?
Summary: The authors bound the search error of non-backtracking dfs of space-partitioning tree (for NN search) with a quantization error.
Interesting paper and while the connection makes sense, I am not convinced that the provided bound is tight, which dilutes the impact of the paper.
Author Feedback

Author rebuttal: We would like to thank the reviewers for their time and feedback. We hope that the ensuing discussion is able to provide some clarification:

Regarding the claims of the paper
* We have tried to qualify throughout the paper that better VQ and larger margins imply better search performance *guarantees*.
* The empirical results show that the actual search performance aligns with the proven performance guarantees.
* MNIST (& Tiny Images) are subsampled because search-error analysis requires brute-force search and is very expensive on large datasets in terms of computation and memory.
* The goal of this paper was to identify factors that affect search performance guarantees. No single method is best on all datasets. Knowing the potential factors affecting search allows us to probe a dataset in question to decide which method to use. For example, the results in this paper indicate that, for a given dataset, the trees with better VQ performance and large margin splits on the dataset might be a better choice for search performance.

Regarding the simplicity of the algorithm analyzed
* Algorithm 1 is used for the clarity of the presentation. For a tree of depth l, this is the first part of the usual backtracking search algorithm where the query reaches the first leaf. The backtracking algorithm continues from here to go back up the tree to look for better candidates.
* The result in this paper can be used to obtain search-time guarantees for the complete backtracking algorithm in the following way -- Thm 3.1 gives the worst-case distance (say d) of the query to the candidate neighbor in first leaf visited. Lemma 3.1 (implicitly) bounds the diameter (say D) of any leaf node. Then the backtracking search algorithm will atmost only consider leaves completely contained in the ball of radius d+D around the query. The only (non-trivial) remaining step is to bound the number of points that might lie in this ball (this would probably depend on the intrinsic dimension). Now better VQ implies better bounds on both d and D (from our results). Hence better VQ implies smaller worst-case ball around the query and hence better worst-case runtime for the backtracking algorithm.

Regarding the comparison of kd-trees to other BSP-trees
* The superiority of the PA-tree/2M-tree/etc over the kd-tree for search time has been empirically exhibited in the cited papers. While the reviewer's observation (that kd-trees require O(1) computation to make split decisions while all other trees require O(dim(X)) time) is right and very valid, in practice, the search time is generally dominated by the number of actual points encountered at the leaves and the total number of leaves visited and not by the node split decisions. This is the reason why the PA-tree/2M-tree/etc have been shown to have better performance than the kd-tree on medium to high dimensional datasets. For kd-tree the (backtracking) search generally ends up traversing all tree leaves.

Regarding the choice of expansion constant over doubling dimension
* The expansion constant is known to be fragile (as indicated by the reviewer's examples). The doubling dimension is a more robust notion of intrinsic dimension. The robustness makes it hard to work with (no search time guarantees are known with the doubling dimension, only known results are with the expansion constant). Ideally we want results with the doubling dimension, but we do not have any yet.
* With the distributional assumption on S and q, it can be shown that the expansion constant is very close to the doubling (or Federer) measure of the distribution, which is more robust that the expansion constant.

Regarding the tightness of the results
* The reviewer's counterexample demonstrates the looseness resulting from considering a single beta* for the whole tree and suggests working on the local level. The goal of this analysis was to present bounds that compare different trees. If psi changes, it changes for all trees. Hence the relative performance of trees still does not change even though the current bound might get looser.
* We considered a single beta* throughout the tree to reduce notation in the result. If we keep track of beta(n) for every node n in the tree, then our current proof will still work and the exp(-l/beta*) term in Eq. 4 will be replaced by exp(-\sum{n in N} (1/beta(n))) where N is the set of nodes seen by Algorithm 1 for the query. This is a tighter result.
* The ratio in lines 531-532 is not necessarily less than 1. An example is when a node contains some (say half the) points which are very closely clustered while the rest are fairly spread apart far away from this cluster and from each other. If a split separates the clustered points from the rest, the ratio can be greater than 1.
* We do not yet have matching lower bounds in addition to the presented upper bounds on the search error.

Regarding clarification questions
* In condition C2, a complete tree of depth l implies a binary tree with all 2^l nodes at depth l.
* Condition C4 is for all nodes at depth l.